Data quantity is more important than its spatial bias for predictive species distribution modelling

Gaul Willson willson.gaul@ucdconnect.ie 1
Sadykova Dinara 2
White Hannah J. 1
Leon-Sanchez Lupe 2
Caplat Paul 2
Emmerson Mark C. 2
Yearsley Jon M. 1
1 School of Biology and Environmental Science, Earth Institute, University College Dublin , Dublin , Ireland
2 School of Biological Sciences, The Queen’s University Belfast , Belfast , United Kingdom
Roberts David
Electronic publication date: 2020 Nov 27
Publication date: 2020
Volume: 8
Electronic Location ID: e10411
Received 2020 Jun 9; Accepted 2020 Nov 2
Copyright: ©2020 Gaul et al.
Copyright year: 2020
Copyright holder: Gaul et al.
License: This is an open access article distributed under the terms of the Creative Commons Attribution License, which permits unrestricted use, distribution, reproduction and adaptation in any medium and for any purpose provided that it is properly attributed. For attribution, the original author(s), title, publication source (PeerJ) and either DOI or URL of the article must be cited.
License URL: https://creativecommons.org/licenses/by/4.0/

Keywords: Biological records, Sample selection bias, Simulation, Spatial bias, Species distribution model, Virtual ecology

Funding: Science Foundation Ireland 15/IA/2881 This publication emanated from research supported by a grant from Science Foundation Ireland under grant number 15/IA/2881. There was no additional external funding received for this study. The funders had no role in study design, data collection and analysis, decision to publish, or preparation of the manuscript.

==============================
Biological records are often the data of choice for training predictive species distribution models (SDMs), but spatial sampling bias is pervasive in biological records data at multiple spatial scales and is thought to impair the performance of SDMs. We simulated presences and absences of virtual species as well as the process of recording these species to evaluate the effect on species distribution model prediction performance of (1) spatial bias in training data, (2) sample size (the average number of observations per species), and (3) the choice of species distribution modelling method. Our approach is novel in quantifying and applying real-world spatial sampling biases to simulated data. Spatial bias in training data decreased species distribution model prediction performance, but sample size and the choice of modelling method were more important than spatial bias in determining the prediction performance of species distribution models.

Introduction

Biological records data (“what, where, when” records of species identity, location, and date of observation) often contain large amounts of data about species occurrences over large spatial areas (Isaac & Pocock, 2015). Knowing the geographic areas occupied by species is important for practical and fundamental research in a variety of disciplines. Epidemiologists use maps of predicted wildlife distributions to identify areas at high risk for wildlife-human transmission (Deka & Morshed, 2018; Redding et al., 2019). Land managers can use knowledge of species distributions in spatial planning to minimize impacts on wildlife of new infrastructure (Dyer et al., 2017; Newson et al., 2017). Because complete population censuses are not available for most species, species distribution models (SDMs) are often used to predict distributions of species using relatively sparse observations of species. Species observation data used to train SDMs must represent the study area, but when studies focus on scales of thousands (or tens- or hundreds of thousands) of square kilometers, it is difficult and often expensive to collect adequate data across the entire study extent. Spatially random or stratified sampling of species across large spatial areas is possible, and such surveys exist for some taxa including butterflies and birds (Uzarski et al., 2017), but such data are uncommon for most taxonomic groups (Isaac et al., 2014). More commonly, data are either spatially extensive but collected opportunistically (Amano, Lamming & Sutherland, 2016), or are collected according to structured study designs but are more spatially limited.

Collecting biological records data is relatively cheap compared to collecting data directly as part of a research project (or at least the costs of collecting biological records are borne in large part by individual observers rather than by data analysts) (Carvell et al., 2016). However, there is an associated challenge because the analyst lacks control over where, when, and how data were collected. Many biases have been documented in biological records data, including temporal, spatial, and taxonomic biases (Boakes et al., 2010). Spatial sampling bias, in which some areas are sampled preferentially, is particularly pervasive at all scales and across taxonomic groups (Amano & Sutherland, 2013; Oliveira et al., 2016). Despite these biases, biological records are often used in species distribution models. Given the ubiquitous presence of spatial sampling bias in biological records data, it is important to know whether spatial bias in training data impedes the ability of SDMs to correctly model species distributions. Data collection efforts often face a practical trade-off between maximizing the overall quantity and the spatial evenness of new records. It would thus be useful to know whether the value of biological records for SDMs can best be improved by increasing the spatial evenness of recording (perhaps at the cost of the overall amount of new data that is added), or by increasing the overall amount of recording (even if new records are spatially biased).

Spatial sampling bias in biological records has similarities with sampling biases that have been investigated in other settings. The field of econometrics uses the term “sample selection bias” to refer to non-random sampling and has developed theory about when sampling bias is likely to bias analyses (Wooldridge, 2009). A key consideration in econometrics’ evaluations of sample selection bias is determining whether the inclusion of data in the sample depends on predictor variables that are included in the model (“exogenous” sample selection), or depends on the value of the response variable (“endogenous” sample selection), or both (Wooldridge, 2009). In ecology, Nakagawa (2015) similarly provides guidelines for assessing missing data in terms of whether data is missing randomly or systematically with respect to other variables (see also Gelman & Hill, 2006). In a machine learning context, Fan et al. (2005) investigated the effect on predictive models of sample selection bias in which sampling is associated with predictor variables—“exogenous sample selection” in the terms of Wooldridge (2009) and “missing at random” in the terms of Nakagawa (2015)—and determined that most predictive models could be sensitive or insensitive to sampling bias depending on particular details of the dataset.

Biological records may have been collected with spatial sampling biases that are exogenous, endogenous, or both, and datasets may contain a mix of records collected with different types of bias. For example, when sampling intensity depends on proximity to roads (Oliveira et al., 2016), the sampling bias is exogenous because records arise from biased sampling that depends on an aspect of environmental space that can be included in models as a predictor variable. However, when a birder, for example, submits a record of an unusual bird from a location where they would not otherwise have submitted records, the bias is endogenous because the sampling location depends on the value of the response variable (species presence). Most sampling biases occur on a continuum and are not unequivocally categorizable using any existing scheme (Nakagawa, 2015), making it difficult to describe exactly the biases in data or predict their effect on model performance.

Studies testing the impact of spatially biased training data on predictive SDMs have shown mixed results (Edwards et al., 2006; Phillips et al., 2009; Barbet-Massin et al., 2012; Stolar & Nielsen, 2015). Phillips et al. (2009) found that spatial bias in the presence records strongly reduced model performance when using a pseudo-absence approach but not when using a presence-absence approach. Using a virtual ecologist simulation approach (Zurell et al., 2010), Thibaud et al. (2014) found that the effect of spatial sampling bias on SDM prediction performance depended on the SDM modelling method, and that the effect of spatial sampling bias was smaller than the effect of other factors, including sample size and choice of modelling method. Warton, Renner & Ramp (2013) provided a method for correcting for spatially biased data when building SDMs, but found that the resulting improvement in model predictive performance was small. Because there is no clear guidance about when spatial bias in training data will or will not affect model predictions, tests of the observed effect of spatial biases common in biological records are important for determining whether those biases are likely to be problematic in practice.

The effect of spatial sampling bias on model predictions can be studied using real or simulated data (Zurell et al., 2010; Meynard, Leroy & Kaplan, 2019). Using real data has the advantage that the biases in the data are, well, real. The spatial pattern, intensity, and correlation of sampling bias with environmental space are exactly of the type that analyses of real data must cope with. However, using real data has two disadvantages. First, the truth about the outcome being modeled (species presence or absence) is not completely known in the real world, making it impossible to evaluate how well models represent the truth. Second, biases in real data are not limited to the biases under study—a study investigating the effect of exogenous spatial sampling bias will be unable to exclude from a real dataset records generated by endogenously biased sampling that depends on the values of the outcome variable. Simulation studies avoid both these problems. Because the investigator specifies the underlying pattern that is subsequently modeled, the truth is known exactly (even when realized instances of the simulation are generated with some stochasticity). The investigator also has direct control over which biases are introduced into a simulated dataset, and therefore can be more confident that any observed effects on predictions are due to the biases under investigation.

Spatial sampling bias can be introduced into simulated data using a parametric function that describes the bias (Isaac et al., 2014; Stolar & Nielsen, 2015; Thibaud et al., 2014; Simmonds et al., 2020) or by following a simplified ad-hoc rule (e.g., splitting the study region into distinct areas that are sampled with different intensities) (Phillips et al., 2009). However, these approaches may not adequately test the effect of spatial bias if the biases found in real biological records do not follow parametric functions or are more severe than artificial parametric or ad-hoc biases. We used observed sampling patterns from Irish biological records to sample simulated species distributions using realistic spatially biased sampling.

We used a virtual ecologist approach (Zurell et al., 2010; Meynard, Leroy & Kaplan, 2019) applied at the scale of Ireland to investigate the effect on the predictive performance of SDMs of (1) spatial sampling bias, (2) sample size (the average number of records per species), and (3) choice of SDM method. We quantified the spatial sampling biases used in our study to enable comparison with biases in other datasets. To the best of our knowledge, our approach is novel in applying real-world spatial sampling biases, derived directly from spatial sampling patterns in existing datasets, to simulated virtual species.

Methods

We assessed the ability of species distribution models to predict “virtual species” distributions (Leroy et al., 2016; Zurell et al., 2010) when the models were trained with datasets with a range of spatial sampling biases and sample sizes. The simulation and analysis process is illustrated in Fig. 1. Virtual species distributions were produced by defining the responses of virtual species to environmental predictor variables (Table 1). We then created maps of “true” virtual species distributions covering 840 10 km × 10 km grid squares in Ireland (total area of study extent = 84,000 km2). We then generated “virtual biological records” by sampling presence-only records from the community of virtual species in each grid square, using sampling patterns taken from real Irish biological records data. We then inferred non-detections for each species using presence records of other species (Van Strien, Van Swaay & Termaat, 2013). SDMs were trained using the environmental variables as predictors and the virtual species detection/non-detection data as the response. Model prediction performance was evaluated using three measures of prediction performance (see ‘Species distribution modeling’).

Figure 1 Simulation and analysis process.

Step 1—define spatial sampling biases: we used the locations of biological records of three different taxa in Ireland to produce rasters of the observed probability of sampling from each 10 × 10 km grid cell in Ireland. Step 2—generate virtual species true distributions: the probability of virtual species occurring in each grid cell was defined as a response to real environmental variables. True presence/absence distributions were generated probabilistically. Step 3—sample virtual species with spatial biases: detection-only observations of species were drawn from sampling locations selected according to the spatial sampling bias rasters generated in Step 1, to create a dataset of virtual detection-only biological records. Step 4—Infer non-detections using detections of other species: non-detections of each species were inferred at locations where other species had been recorded, transforming the detection-only virtual biological records into detection/non-detection data. Step 5—train SDMs: SDMs were trained with the virtual detection/non-detection biological records as the response variable and the real environmental variables as predictors. Step 6—test SDMs: SDM prediction performance was assessed using three metrics, which capture different aspects of models prediction performance. Steps 5 and 6 (blue bounding box) were conducted using a spatial block cross-validation framework illustrated in Fig. 3.

Table 1 Environmental predictor variables used to define and model the distribution of virtual species in Ireland.

Moran’s I values indicate the spatial clustering of values for each variable, where a value of one indicates strong spatial clustering of variable values, zero indicates random spatial arrangement of values, and negative one indicates strongly dispersed spatial arrangement of values. Details of data sources are in the Methods section.

Variable	Description	Data Source	Moran’s I	
Annual minimum temperature (degrees C)	2% quantile of annual temperatures in each grid cell averaged over the years 1995-2016	E-OBS	0.84	
Annual maximum temperature (degrees C)	98% quantile of annual temperatures in each grid cell averaged over the years 1995-2016	E-OBS	0.83	
Annual precipitation (mm)	Average total annual precipitation in each grid cell over the years 1995-2016 (excluding 2010-2012)	E-OBS	0.82	
Average daily sea level atmospheric pressure (hecto Pascals)	Average daily sea level atmospheric pressure over the years 1995-2016	E-OBS	0.86	
Agricultural areas	Proportion of each grid cell classified as agricultural areas	CORINE Land Cover Database	0.53	
Artificial surfaces	Proportion of each grid cell classified as artificial surfaces	CORINE Land Cover Database	0.44	
Forest and semi-natural areas	Proportion of each grid cell classified as forest and semi-natural areas	CORINE Land Cover Database	0.41	
Water bodies	Proportion of each grid cell classified as water bodies	CORINE Land Cover Database	0.35	
Wetlands	Proportion of each grid cell classified as wetlands	CORINE Land Cover Database	0.55	
Elevation	Average elevation in each grid cell	ETOPO1	0.29	

Environmental predictor variables

We chose environmental predictor variables with a range of spatial patterns and scales of spatial auto-correlation (Table 1, Fig. S1). We used real environmental variables measured over a real geographic space, so the variety of spatial patterns in our predictor variables should be similar to patterns in variables that determine biological species distributions at this scale, adding realism to our simulation. We used climate variables (which show relatively strong spatial clustering, Table 1) from the E-OBS European Climate Assessment and Dataset EU project (Haylock et al., 2008; Van den Besselaar et al., 2011). We calculated the proportion of each grid square covered by different land cover variables (which show less spatial clustering than climate variables, Table 1) from the CORINE Land Cover database (CORINE land cover database, 2012). We calculated the average elevation within each grid square by interpolation using ordinary kriging from the ETOPO1 Global Relief Model (Amante & Eakins, 2009).

Spatial data were prepared using the ‘sf’, ‘sp’, ‘raster’, ‘fasterize’, ‘rgdal’, ‘gstat’, and ‘tidyverse’ packages in R version 3.6 (Bivand, Keitt, & Rowlingson, 2018; Gräler, Pebesma & Heuvelink, 2016; Hijmans, 2018; Pebesma, 2018; R Core Team, 2020; Ross, 2018; Wickham, 2017).

Species presence data

We downloaded observations of species across the island of Ireland for the years 1970 to 2014 from the British Bryological Society for bryophytes and from the Irish National Biodiversity Data Centre (NBDC) for moths, butterflies, and birds. We used data for taxa that varied in both their popularity with recorders and the ease with which species can be identified, because we expected this would translate into different spatial sampling biases in the data. The data contained presence-only records of species, with the date and location of the observation, and an anonymized observer identifier. Locations of records were provided as either 1 km2 or 100 km2 (10 km × 10 km) grid squares, but we used 10 km × 10 km grid squares in all analyses in order to retain the majority of the data. Within each taxonomic group, we grouped records into sampling event checklists, where a sampling event was defined as all records with an identical combination of recording date, location, and observer.

Spatial sampling patterns in Irish species presence data

For each taxonomic group, we quantified sampling effort in each grid square as the proportion of all records coming from the grid square. We quantified spatial sampling bias by calculating the spatial evenness of sampling effort among locations using Simpson evenness (Magurran & McGill, 2011).

Data simulation

Simulating species distributions

We simulated and sampled virtual species distributions using the ‘virtualspecies’ package (Leroy et al., 2016) in R. For each virtual species, seven environmental variables (Table 1) were randomly chosen to use as drivers of occurrence (only seven of the ten variables shown in Table 1 were used for each species so that not all species responded to exactly the same environmental variables). The seven selected environmental variables were centered, scaled, and summarized using principal components analysis with the ‘ade4’ R package (Dray & Dufour, 2007). The probability of occurrence of each virtual species i in each grid square j was a logistic function of the first two principal components and their quadratic terms: logitpij=αi+ ∑k=12β1kiVkj+β2kiVkj2

where pij is the probability that virtual species i occurs in grid square j, Vkj indicates the value of the kth principal component in grid cell j, and the α and β terms are the species-specific coefficients defining the response of the virtual species to the environment. The coefficients for each virtual species (α, β1k, β2k) were chosen to ensure that each virtual species was present in at least eight of the 840 grid squares.

Realized species distributions

A single realized distribution of each virtual species i was created by randomly generating a “presence” (1) or “absence” (0) for each grid square j by drawing a value from a binomial distribution with probability pij. We simulated a community containing 1,268 virtual species (the number of recorded bryophyte species in Ireland). For comparison, results of a simulation using a small community of 34 species are in Article S1.

Sampling realized species with spatial bias

Virtual biological records data were generated by sampling the realized species communities in “sampling events” at different locations to produce spatially explicit species checklists (Fig. S2). Spatial sampling locations were chosen based on spatial sampling patterns from three Irish biological records datasets with different spatial sampling biases (Table 2): birds (low spatial sampling bias), butterflies (median spatial sampling bias), and moths (severe spatial sampling bias). This gave four spatial sampling “templates”, including the case of no spatial sampling bias (Fig. 2).

Table 2 Spatial sampling evenness of the spatial sampling template datasets measured using Simpson evenness.

A value of one indicates perfectly even sampling (all grid squares containing the same number of records). Lower Simpson evenness values indicate more spatially uneven sampling.

Spatial sampling template	Simpson evenness value	
No bias	1	
Low bias (birds)	0.762	
Median bias (butterflies)	0.126	
Severe bias (moths)	0.021	

Figure 2 Spatial sampling patterns from Irish biological records.

Spatial sampling patterns from Irish biological records were used as templates to create virtual species records data with varying amounts of spatial bias. Darker shades indicate higher relative probability of sampling from a grid square compared to other grid squares within the same template; overall sampling effort is the same for each panel (A) through (D). The most heavily sampled grid square in each spatial bias template has a relative recording effort of one, while a grid square with half as many records as the most heavily sampled square has a relative recording effort of 0.5. Spatial sampling patterns derived from datasets for different taxonomic groups were: (A) no bias (even probability of sampling from every grid square), (B) low bias (based on bird data), (C) median bias (based on butterflies), and (D) severe bias (based on moths).

To make sampling patterns comparable between datasets with different sample sizes, we calculated a relative sampling weight for each grid square in each empirical dataset (where the most heavily sampled cell had a weight of one) by counting the number of records in each grid square and dividing by the maximum number of records in any grid square (Fig. 2).

We created virtual biological records with six different sample sizes, defined as the mean number of records per species (number of records per species = 2, 5, 10, 50, 100, and 200).

To generate virtual biological records from the virtual species communities, we randomly selected a grid square, using selection probabilities from one of the four spatial-bias templates. Within each grid square that was selected for sampling, we (1) generated a list of virtual species that were present in the grid square; (2) defined the probability of observing each of the present species based on the species’ prevalence in the entire study extent (so that common species had a higher probability of being recorded when present), and (3) drew presence observations with replacement from the list of present species. We continued this sampling process until we had accumulated the desired number of records.

Species distribution modeling

We tested three different SDM modeling techniques: generalized linear models (GLMs) (Hosmer & Lemeshow, 2000), boosted regression trees (Elith, Leathwick & Hastie, 2008; Friedman, 2001), and inverse distance-weighted interpolation (Cressie, 1991). These represent distinct types of methods used for SDMs, including linear and machine learning methods, and a spatial interpolation method that does not include information from environmental covariates. For all methods, the modeled quantity was the probability of the focal virtual species being recorded on a checklist. We modeled each species individually as a function of five environmental predictor variables, chosen from the ten possible predictor variables listed in Table 1. Using only five of the ten possible predictor variables simulated a real-world situation in which the factors that influence species distributions are not entirely known. We treated the list of records from each sampling event as a complete record of that sampling event, and treated the absence of species from a sampling event checklist as non-detection data for those species (Fig. S2; Van Strien, Van Swaay & Termaat, 2013; Kéry et al., 2010). Thus, we explicitly used a detection/non-detection rather than a presence-only modeling framework. Our approach of inferring non-detections of species at the locations of presence records of other species in the community is similar to the “target-group background” approach of Phillips et al. (2009). Using non-detection data inferred from records of other species ensured that the sampling biases were the same for detections and non-detections, which may reduce the effect of sampling bias (Barbet-Massin et al., 2012; Johnston et al., 2020; Phillips et al., 2009).

We modeled 110 randomly selected virtual species from the 1,268 virtual species in the large community simulation. The number of virtual species modeled was a compromise between high replication and computation limitations, but testing 110 virtual species should provide enough replication for robust conclusions. We fitted each type of SDM once to each combination of virtual species, sample size, and spatial sampling bias. Thus, the sample size for our study—the number of SDM prediction performance values that we used to assess the effects of spatial sampling bias, sample size, and SDM method—was 110 prediction performance values for each combination of SDM method, sample size, and spatial sampling bias. Replication in our study came not from repeatedly fitting models to different randomly generated sets of presences and absences of the same virtual species, but rather from fitting each model once to data for many different virtual species, all generated using parameters randomly drawn from the same distributions. However, the same 110 virtual species were used for each combination of SDM method, spatial sampling bias, and sample size, ensuring that all comparisons were based on the same underlying task (i.e., modelling the same true species distributions).

Models were trained and evaluated using five-fold spatial block cross-validation (Roberts et al., 2017) that partitioned the study extent into spatial blocks of 100 km × 100 km and allocated each block to one of five cross-validation partitions. We only attempted to fit models if there were more than five presence records in the training data. Prediction performance of models was evaluated using the true virtual species presence or absence in each grid cell not included in the spatial extent of the training partitions (Fig. 3). Thus, evaluation data was spatially even and the number of evaluation points stayed constant even as the sample size and spatial bias of training data changed (Fig. 3). Prediction performance was evaluated using three metrics. The area under the receiver operating characteristic curve (AUC) (Hosmer & Lemeshow, 2000) evaluated models’ ability to accurately rank locations where species were present or absent using the continuous SDM predictions, Cohen’s Kappa (Cohen, 1960), calculated using the threshold that maximised Kappa, evaluated models ability to transform continuous SDM predictions into binary maps of presence and absence, and root mean squared error (RMSE) evaluated model calibration.

Figure 3 Species distribution model training and testing process for a single cross-validation fold.

The true virtual species distribution (A, presences shown in dark green, absences in light grey) was sampled to produce virtual biological records with a range of sample sizes and spatial biases, including no bias (B) and median bias (C). Orange points in (B) and (C) show checklists on which the species was recorded, black points show checklists on which the species was not recorded (i.e., non-detection points). Species distribution models were fit using five-fold spatial block cross validation, in which data from about 80% of the spatial area was used to train models (light grey background in B and C). Data from the remaining spatial areas (dark grey background in B and C) was set aside for model evaluation. Model evaluation tested the ability of species distribution models to predict the true presence (orange dots) or absence (black dots) of the species in each grid cell within the evaluation areas (D). Model evaluation therefore used spatially even data with the same number of evaluation points (D) regardless of the sample size and spatial bias of training data (B and C).

To train GLM SDMs, we used logistic regression (‘glm’ function) with a binomial error distribution and logit link. Quadratic terms were fitted, but we did not fit interactions between variables. We controlled overfitting by limiting the number of terms in GLMs such that there were at least 10 detections or non-detections (whichever was smaller) in the training data for each non-intercept term in the model. If a quadratic term was included in a model, we also included the 1st degree term. For generating predictions, we used the model that gave the lowest AIC based on the training data.

Boosted regression trees were trained using ‘gbm.step’ in the ‘dismo’ package (Greenwell, Boehmke & Cunningham, 2018; Hijmans et al., 2017). Unlike GLMs, boosted regression trees do not require the modeler to specify interactions between variables, because the trees will discover and model interactions if they are present. The tree complexity specified by the modeler controls the maximum interaction order that the models are permitted to fit, and therefore can be used to prevent overfitting. We tested tree complexities of two and five, to build models that allowed interactions between up to two and up to five variables, respectively. We used learning rates small enough to grow at least 1,000 trees (following Elith, Leathwick & Hastie, 2008), but large enough to keep models below an upper limit of 30,000 trees because of computation time limitations. We used gbm.step to determine the optimal number of trees for each model, based on monitoring the change in 10-fold cross-validated error rate as trees were added to the model (Hijmans et al., 2017). Limiting models to a maximum of 30,000 trees did not affect our results. Details of the procedure are in Article S1 and in our R code (see Data and code accessibility statement).

Inverse distance-weighted interpolation was implemented using ‘gstat’ (Gräler, Pebesma & Heuvelink, 2016; Pebesma, 2004). We tuned parameters of the inverse distance-weighted interpolation model based on prediction error (details in Article S1).

After models were fitted, we looked for evidence of overfitting and assessed whether the number of presence records of the focal species in the test dataset affected prediction performance metrics (details in Article S1). All analyses used R version 3.6.0 (R Core Team, 2020).

Analyzing effects of sampling bias and sample size

We modeled the predictive performance (AUC, maximum Cohen’s Kappa, and RMSE) of SDMs as a function of spatial sampling bias, sample size (average number of observations per species), and SDM method. Modelling was done using boosted regression trees (‘gbm.step’ in the ‘dismo’ package) (Greenwell, Boehmke & Cunningham, 2018; Hijmans et al., 2017). To assess whether species prevalence (the commonness or rarity of a species in the study extent) affected our evaluations of model performance, we graphed prediction performance as a function of species prevalence for all models (Fig. S3). We included species prevalence in the boosted regression tree models of RMSE, but not in our analysis of AUC or Cohen’s Kappa, because only RMSE showed a strong trend with species prevalence (Fig. S3). Variable importance was assessed based on the reduction in squared error attributed to each variable in boosted regression tree models (Friedman, 2001). We also assessed the effect of spatial sampling bias and sample size of training data on the number of species for which models could be fitted within the computational time and memory constraints of this study (Article S1).

Results

Simulated species showed a variety of plausible distribution patterns (Fig. 4) and prevalences (Fig. S4), including species with north/south distribution gradients and distributions that followed geographic features such as the coastline (Fig. 4).

Figure 4 The true distributions of four example simulated species.

Simulated species showed a range of plausible distributions with a range of prevalences, including (A) common widespread species, (B) rare species mostly limited to north-western coastal sites, (C) species with a north/south gradient in occurrence, and (D) common species that are absent from southern sites.

Sample size (the mean number of observations per species) and choice of SDM method were the most important variables for explaining variations in prediction performance of SDMs (Table 3). Spatial sampling bias was the least important variable for explaining variation in prediction performance for all three performance metrics (Table 3). Simpson evenness values for spatial sampling evenness of the template datasets are in Table 2.

Table 3 Importance of sample size, spatial bias, modelling method, and species prevalence for determining predictive performance of species distribution models.

Variable importance measures from a boosted regression tree show the relative influence of sample size (average number of records per species), species distribution modeling method, and spatial bias in training data on three different measures of prediction performance of species distribution models. The relative influence for each variable is the reduction in squared error attributed to that variable in a boosted regression tree model. Prediction performance metrics were area under the receiver operating characteristic curve (AUC), root mean squared error (RMSE) and Cohens, Kappa calculated using the threshold that maximized Kappa (Kappa). The effect of species prevalence was not included for AUC and for Kappa, because exploratory plots showed no indication of an effect.

Prediction performance metric	Variable	Relative importance (reduction in squared error)	
AUC	Average number of records per species	78.5	
	Species distribution modelling method	14.8	
	Spatial bias	6.7	
RMSE	Species prevalence	99.9	
	Average number of records per species	0.2	
	Species distribution modelling method	0.003	
	Spatial bias	0.0001	
Kappa	Species distribution modelling method	74.2	
	Average number of records per species	22.4	
	Spatial bias	3.4	

Number of species successfully modeled

The number of species for which models fitted successfully increased as sample size increased and spatial bias decreased (Fig. 5). For GLMs and inverse distance-weighted interpolation, model fitting was largely successful when datasets had more than 100 records per species, except when spatial bias was severe (Fig. 5). Boosted regression trees failed to fit models for some species even with relatively large amounts of data (e.g., an average 200 records per species), and models fit less frequently when data had median or severe spatial biases (Fig. 5). The effect of spatial bias on the number of species for which models fitted was small, but was slightly greater for boosted regression trees than for other SDM modelling methods (Fig. 5).

Figure 5 The number of virtual species successfully modeled.

The number of virtual species (out of 110 total species chosen for modelling from the large community simulation) for which species distribution models fitted within the computation time and memory constraints we imposed, according to the spatial sampling bias and sample size of training data and the species distribution modelling method. Species distribution modelling methods were (A) generalized linear models, (B) boosted regression trees, and (C) inverse distance-weighted interpolation. Spatial biases were no bias (Simpson evenness = 1), low (e.g., birds, Simpson evenness =0.76), median (e.g., butterflies, Simpson evenness =0.13), and severe (e.g., moths, Simpson evenness =0.02).

Predictive performance of SDMs

The amount of spatial bias in training data was less important than sample size and choice of SDM method in predicting the performance of SDMs (Table 3, Table S1). AUC for predictive SDMs increased with the average number of records per species and with decreasing spatial bias in the training data when using all SDM methods (Figs. 6 and 7). Root mean squared error (RMSE) was largely unaffected by spatial sampling bias (Fig. 8, Table 3). Cohen’s Kappa differed between SDM methods, but generally increased with sample size, and with decreasing spatial bias in the training data (Figs. S5 and S6). However, Cohen’s Kappa was low for all methods and combinations of sample size and spatial bias when evaluated using spatial block cross-validation (Fig. S6). The low Kappa scores indicated that our models were unable to generalize beyond the training data when the task was to produce binary maps, despite the fact that the models retained the ability to generalize when the task was to rank sites according to the continuous SDM outputs (Article S1, Fig. S7). Species prevalence (the number of grid squares occupied by a species) had a negligible effect on the average value of AUC, though it did affect the variability of AUC (Fig. S3). Species prevalence strongly affected the expected value of RMSE, with RMSE increasing with species prevalence (Table 3, Fig. S3).

Figure 6 Expected prediction performance of species distribution models for 110 simulated species under a range of sample size and spatial sampling bias scenarios.

Panels show the expected prediction performance of species distribution models constructed using (A) generalize linear models, (B) boosted regression trees, and (C) inverse distance-weighted interpolation. Lines show expected area under the receiver operating characteristic curve (AUC) given the sample size and spatial sampling bias of training data, and the species distribution modelling method. Rug plots indicate sample sizes (mean number of records per species) of the virtual biological records datasets used to train species distribution models.

Figure 7 Observed prediction performance (AUC) of species distribution models for 110 virtual species under a range of sample size and spatial sampling bias scenarios.

Panels show the observed area under the receiver operating characteristic curve (AUC) of species distribution models constructed using (A) generalized linear models, (B) boosted regression trees, and (C) inverse distance-weighted interpolation. Boxes contain the middle 50% of the observed AUC values. The horizontal line within each box indicates the median AUC value. Each box plot (box, whiskers, and outlying points) represents 110 observations (one for each virtual species) unless models failed to fit for some species (see Fig. 4). The width of boxes is proportional to the square root of the number of observations in that group.

Figure 8 Observed prediction performance (RMSE) of species distribution models for 110 virtual species under a range of sample size and spatial sampling bias scenarios.

Panels show the observed root mean squared error (RMSE) of species distribution models constructed using (A) generalized linear models, (B) boosted regression trees, and (C) inverse distance-weighted interpolation. Boxes contain the middle 50% of the observed RMSE values. The horizontal line within each box indicates the median RMSE value. Each box plot (box, whiskers, and outlying points) represents 110 observations (one for each virtual species) unless models failed to fit for some species (see Fig. 4). The width of boxes is proportional to the square root of the number of observations in that group.

Effect of sample size

Sample size (average number of records per species) was the most important variable for predicting species distribution model prediction performance in terms of AUC (Table 3). AUC improved with increasing average number of records per species for all SDM methods, and the improvement in AUC decelerated as the number of records per species increased (Figs. 6 and 9). Kappa improved with increasing average number of records per species for two of the three SDM methods, and the improvement in Kappa decelerated as the number of records per species increased (Fig. S5).

Figure 9 Contour plot of expected prediction performance of species distribution models as a function of the sample size and spatial sampling bias in virtual biological records datasets.

Expected prediction performance (AUC, contours and shading) of generalized linear model (GLM) species distribution models for virtual species, according to the spatial sampling evenness and sample size of training data. Note the different scales of the horizontal axes in A and B. (A) shows detail of prediction performance changes when sample sizes were small. (B) shows larger sample sizes. Spatial sampling evenness was quantified using Simpson evenness. High values of Simpson evenness indicate minimal spatial bias. Open circles show the values of sample size and spatial sampling evenness for virtual biological records datasets used to train species distribution models. Filled black circles show sample size and spatial sampling evenness of Irish biological records datasets used as spatial sampling templates.

Effect of spatial bias

Higher levels of spatial sampling bias generally reduced AUC and Kappa, but the size of this effect was small for the low level of bias (Fig. 6). SDMs built with GLMs showed the biggest difference in prediction performance between models trained with unbiased data and models trained with data showing median spatial bias (reduction in expected AUC of 0.037 when using an average of 200 records per species, Fig. 6). Other SDM methods showed less difference in AUC between models trained with unbiased data and models trained with data containing median spatial bias (decrease in expected AUC of 0.033 for boosted regression trees and 0.030 for inverse distance-weighted interpolation when using an average of 200 records per species).

The AUC for inverse distance-weighted interpolation models trained with unbiased data was generally higher than the AUC for GLMs and boosted regression trees trained with severely biased data, but lower than the AUC for GLMs and boosted regression trees trained with data with median spatial bias for any given sample size (Figs. 6 and 7).

Discussion

Both sample size (the average number of observations per species) and choice of modelling method were more important than the spatial bias of training data for determining model prediction performance. This is in line with the results of Thibaud et al. (2014), and the importance of sample size was in line with studies summarized in Meynard, Leroy & Kaplan (2019). Thibaud et al. (2014) simulated spatial sampling bias by defining sampling probability as a linear function of distance from the nearest road. In contrast, our study used observed spatial sampling patterns from real biological records datasets. Our results therefore provide a more direct confirmation that spatial biases of the type and intensity found in real datasets are not as important as other factors in determining SDM prediction performance. We did not measure how different our biased sampling was from Thibaud et al. (2014) biased sampling, but we suspect biases in real data are more severe than the bias tested by Thibaud et al. (2014). More attempts to quantify sampling biases in real data would be valuable for informing how bias is incorporated into future simulation studies.

While spatial bias was not the most important factor determining SDM prediction performance, spatial sampling bias did affect model prediction performance when spatial bias was relatively strong. The limited effect of spatial bias on SDMs that we observed is similar to other findings that have shown spatial sampling bias to have a small effect on model performance (Thibaud et al., 2014; Warton, Renner & Ramp, 2013) or to affect only some SDM methods (Barbet-Massin et al., 2012). Our study used spatial biases and the spatially explicit environmental data representative of data likely to be used in SDMs using biological records in Ireland. Our conclusions therefore apply most directly to applications of SDMs using Irish biological records, and may not be generalizable to other geographic locations, or for species within Ireland that do not respond to the environmental predictor variables used in this study. Our simulations could be “scaled up” to use environmental variables from, and create virtual species over, a larger spatial extent, which would provide insight about whether spatial sampling bias remains relatively un-important at larger (e.g., continental) scales. However, our results strengthen a growing body of literature that suggests that spatial sampling bias is rarely the most important issue in determining SDM prediction performance. In particular, the choice of modelling method may often have more impact on SDM prediction performance than a variety of other factors (Barbet-Massin et al., 2012; Fernandes, Scherrer & Guisan, 2018).

Training data with low spatial sampling bias produced species distribution models that performed nearly as well as models trained with unbiased data, when the task was ranking the probability of a species being detected at sites. Prediction performance was poor when models were trained with small sample sizes, regardless of the spatial bias in training data. Similarly, model performance increased quickly with sample size when sample size was small, even when the data had severe spatial bias. This suggests that, for taxonomic groups with relatively few records per species, the usefulness of the data for predictive SDMs can be improved by increasing sample size, even if additional data collection is spatially biased. In contrast, for taxonomic groups for which biological records datasets already have a high average number of records per species (e.g., birds and butterflies which both have an average of over 2,000 records per species in Ireland) further improvements in SDM prediction performance will likely require increasing the spatial evenness of data (Fig. 9).

While our SDMs were able to correctly rank sites in terms of the probability of a focal species being detected (measured using AUC), the SDMs were essentially unable to convert the continuous SDM outputs into binary presence/absence maps (measured using Cohen’s Kappa) for locations outside the training data (Fig. S6). Converting continuous SDM outputs to binary classifications is fraught with difficulties, and the criteria that should be used for selecting the threshold for conversion depend on the purpose of the SDMs (Guillera-Arriota et al., 2015, and references therein). Any conversion of a continuous SDM output to a binary classification necessarily results in a loss of information, and Guillera-Arriota et al. (2015) suggest that for most applications of SDMs, binary conversions are not necessary. It was outside the scope of this study to explore why SDMs were better at ranking than at producing binary classifications for sites outside the training data. The main finding of our study, that spatial sampling bias was less important than other factors in determining model performance, was consistent regardless of whether prediction performance was evaluated in terms of the ability to rank (AUC) or classify (Kappa) sites (Table 3).

Presence-only data are common in biological records, but there are very few true “presence-only” SDM modeling methods. Most common SDM methods require information on presences and something else (e.g., non-detections, or background- , quadrature-, or pseudo-absences points) (Phillips et al., 2009; Elith et al., 2011; Renner et al., 2015). We inferred non-detections from detections of other species (e.g., Phillips et al., 2009; Van Strien et al., 2010; Isaac et al., 2014; Johnston et al., 2020). Other options include generating pseudo-absences randomly or according to a variety of rules (Barbet-Massin et al., 2012). Phillips et al. (2009) proposed a “target group” approach that selects background points using detections of other species. The target group approach is essentially identical to ours. Our results—particularly the finding that spatial sampling bias was relatively un-important relative to other factors—applies most directly to SDM approaches that infer non-detections from presence records of other species. When background or pseudo-absence points are generated randomly or using an ad-hoc rule, we expect that SDMs will be more strongly impacted by spatial bias in the presence-only data than were SDMs in our study (Phillips et al., 2009).

The objective of our SDMs was to fill in gaps in species distribution knowledge within the spatial and environmental conditions of the island of Ireland, an area of about 84,000 km2. Our results may not generalize to larger spatial scales or to cases in which the goal of SDMs is uncovering species’ entire fundamental environmental niche or determining the environmental factors most strongly influencing distributions. The spatial scope of our SDMs is sensible both from an ecological and applied standpoint, because the island of Ireland is a geographically delimited ecological unit, and because decision making about species conservation and management often happens within political units (e.g., nations, states, or counties) that cover only a portion of species’ spatial and environmental distributions.

GLMs had the best prediction performance of the SDM methods we tested, even though they were more affected by spatial bias than were other methods. The high performance of GLMs relative to other modelling methods in this study agrees with the simulation results of Thibaud et al. (2014) and Fernandes, Scherrer & Guisan (2018). However, as in both those studies, we generated virtual species distributions according to a linear model, so it is possible that the good performance of GLMs is due to the model having the same functional form as the “true” species responses. In real applications, it is unlikely that the functional form of the model will exactly match the form of the true species responses. Indeed, the species distribution modelling literature has many examples of different modelling methods performing best in different studies, suggesting that no modelling method consistently outperforms others (Bahn & McGill, 2007; Breiner et al., 2018; Cutler et al., 2007; Elith et al., 2006; Elith & Graham, 2009).

Boosted regression trees’ prediction performance was slightly less affected by spatial bias than GLMs’, and prediction performance of both methods was similar when trained with large, spatially biased datasets. But boosted regression trees failed to fit models more often than did GLMs, especially when sample sizes were smaller, which may make them inferior to other modelling methods for small datasets, at least within the computational resource limits we faced. We cannot rule out the possibility that the performance of boosted regression trees would improve if they were trained with a smaller learning rate and permitted to grow more than 30,000 trees. However, most users of SDMs will face some computational resource limitations.

In this study, we introduced spatial bias specifically into the training data and tested model performance using spatially even evaluation data. However, spatial bias can also occur in evaluation data and may affect the reliability of model evaluations (Fink et al., 2010). When using real biological records datasets, it is likely that both model training and evaluation will use spatially biased data, making it difficult to dis-entangle whether observed effects of spatially biased data on prediction performance are due to the influence of biased data in the model training step or in the model evaluation step. We evaluated models on spatially even data (which is easy using simulated data but would be more difficult or impossible when using real data), so the observed effects of spatially biased data on prediction performance in our study can be attributed to the effect of biased data on model training. All of the SDM methods we used involve some kind of model evaluation as part of the model training process, either inherent in the model fitting or introduced by our implementation. For example, with our GLMs we introduced a model evaluation step to select predictor variables. The final GLM models were therefore based on variables that had been selected by evaluation on spatially biased data. For both GLMs and inverse distance-weighted interpolation, it is possible that using unbiased data in the evaluations during model selection would have led to different final models. Therefore, the observed effect of the spatial bias in this study could be due to how biased data affects the actual fitting of each individual model, or to how the biased data affects the evaluation step used to select which fitted model to use for predictions. Tree-based methods, including boosted regression trees, select which values of predictor variables to split at and/or which predictor variables to use at each node based on how much those splits improve some measure of performance on the training data (Elith, Leathwick & Hastie, 2008; Hastie, Tibshirani & Friedman, 2009). Thus, evaluation on potentially spatially biased training data is inherent in fitting tree models.

Fink et al. (2010) provided a method for correcting spatial bias in evaluation data to reduce the effect of spatial bias on model evaluation, but they did not explicitly address spatially biased data in model training. Our results showed that spatially biased data can impact model training (at least when the spatial bias is relatively strong). Investigating the effect of spatially biased data on the evaluation that takes place as part of model training (e.g., during variable selection or parameter tuning) may be a worthwhile path for future research. It may be possible to use a method like that proposed by Fink et al. (2010) to correct spatial bias during the evaluation that takes place within the model training process.

Conclusion

We found that spatial sampling bias in training data affected species distribution model prediction performance when the spatial bias was relatively strong, but that sample size and the choice of modelling method were more important than spatial bias in determining model prediction performance. This study adds to a body of literature suggesting that prediction performance of species distribution models is less affected by spatial sampling bias in training data than by other factors including modelling method and sample size. We suggest that, when biological records datasets are relatively small, model prediction performance can best be improved by increasing the number of records, even if additional data are sampled with spatial bias. Attempts to reduce spatial bias in data through data filtering should be cautious about the resulting decrease in sample size, which could cancel any gains in prediction performance from reducing spatial sampling bias. Converting continuous SDM outputs to binary maps should be done with caution, and prediction performance tests should use spatial block cross-validation or test on independent data.

Supplemental Information

Figure S1 Mapped values of environmental variables used to define and model virtual species distributions in Ireland

Variables were chosen to represent a range of spatial patterns and scales of spatial auto-correlation. Units of measurement for variables are: degrees Celcius (minimum temperature and maximum temperature); millimeters (annual precipitation); hecto Pascals (atmospheric pressure); proportion of grid square covered by land cover (agricultural areas, artificial surfaces, forest & semi-natural, wetlands, water bodies); meters (elevation).

Click here for additional data file.

Article S1 Supplementary Methods and Results

Click here for additional data file.

Figure S2 Example virtual species occurrence checklist

We simulated an observation process that produces “presence only” records of a list of species at a location (A), which is a common format for biological records data. We then inferred species non-detections (represented as “0” in B) for every species in the community. This produced complete detection/non-detection data for every species on every checklist (B). We used the detection/non-detection data (B) to train species distribution models.

Click here for additional data file.

Figure S3 Prediction performance measures of species distribution models for virtual species (large community simulation) according to species prevalence

Prevalence is defined as the proportion of grid cells in which the species occurred. RMSE increased with species prevalence. AUC and Kappa did not show a directional trend with prevalence, but the variability in AUC is higher for low-prevalence species.

Click here for additional data file.

Figure S4 Distribution of prevalences of simulated virtual species

Prevalence is defined as the proportion of grid cells in which the species occurs. There were many relatively rare species (present in few grid cells) and few common species (present in many grid cells). We did not keep virtual species with prevalences lower than 0.01 (equivalent to the virtual species occurring in at least eight of the 840 grid squares in our study extent) because we expected such species to be poor candidates for species distribution modelling.

Click here for additional data file.

Table S1 Adjusted R2 of generalized additive models (GAMs) modeling prediction performance (AUC) of species distribution models for simulated species

The full model (1st line) models AUC as a function of spatial bias and a smooth of sample size by spatial bias. The 2nd and 3rd lines show the the adjusted R2 of models with the sample size or spatial bias terms removed, respectively. Removing the sample size term reduces the adjusted R2 more than removing the spatial bias term does.

Click here for additional data file.

Figure S5 Expected prediction performance (Cohen’s Kappa) of species distribution models for 110 simulated species under a range of sample size and spatial sampling bias scenarios

Panels show the expected Cohen’s Kappa for species distribution models constructed using (A) generalized linear models, (B) boosted regression trees, and (C) inverse distance-weighted interpolation. Lines show expected Kappa given the sample size and spatial sampling bias of training data, and the species distribution modelling method, when the threshold for binary conversions was selected by maximising Kappa. Rug plots indicate sample sizes (mean number of records per species) of the virtual biological records datasets used to train species distribution models.

Click here for additional data file.

Figure S6 Observed prediction performance (Cohen’s Kappa) of species distribution models for 110 virtual species under a range of sample size and spatial sampling bias scenarios

Panels show the observed value of Cohen’s Kappa evaluated on the training data (A, C, E) and using spatial block cross-validation (B, D, F) for species distribution models constructed using (A, B) generalized linear models, (C, D) boosted regression trees, and (E, F) inverse distance-weighted interpolation. Boxes contain the middle 50% of the observed Kappa values. The horizontal line within each box indicates the median value. Each box plot (box, whiskers, and outlying points) represents 110 observations (one for each virtual species) unless models failed to fit for some species. The width of boxes is proportional to the square root of the number of observations in that group.

Click here for additional data file.

Figure S7 Generalized additive model predictions of discrimination performance of species distribution models for simulated species evaluated with and without spatial block cross-validation

Prediction performance (AUC) was higher when performance was measured on training data (i.e. without cross-validation) than when performance was measured using spatial block cross-validation. Lines show the expected prediction performance of species distribution models (SDMs) using (A) GLMs, (B) boosted regression trees, and (C) inverse distance-weighted interpolation. Lines show the predicted area under the receiver operating characteristic curve (AUC) from GAM models. Rug plots indicate sample sizes (in terms of mean number of observations per species) of the simulated datasets that were used to train SDMs.

Click here for additional data file.

Figure S8 Number of terms used by generalized linear model (GLM) species distribution models as a function of sample size and spatial sampling bias

The number of terms selected by automated forward and backward stepwise variable selection for logistic regression species distribution models increased as sample size (average number of records per species) increased. Boxplots show the distribution of the number of terms per model at the different sample sizes tested. Lines show the loess-smoothed number of terms selected as a function of sample size for four different levels of simulated spatial sampling bias.

Click here for additional data file.

Figure S9 Prediction performance of generalized linear model (GLM) species distribution models as a function of the number of terms in the model for models trained with different amounts of data

The prediction performance (measured using AUC) was generally lowest when fewer terms were used in GLM species distribution models. Panels show results for species distribution models trained with sample sizes of an average of (A) 2, (B) 5, (C) 10, (D) 50, (E) 100, and (F) 200 observations per species. Box plots show the distribution of AUC values for models with different numbers of terms.

Click here for additional data file.

Figure S10 Number of trees used to fit boosted regression tree species distribution models when training data had different levels of spatial sampling bias and different sample sizes

Box plots show the distribution of the number of trees used to fit boosted regression tree species distribution models when training data had different sample sizes from two to 200 observations per species, and different simulated spatial sampling biases from no bias to severe bias. The width of the box plots is proportional to the square root of the number of observations in each group. Note the logarithmic scale of the horizontal axis. Most models used fewer than 10,000 trees.

Click here for additional data file.

Figure S11 Prediction performance of boosted regression tree species distribution models as a function of the number of trees used

The graph shows narrow box plots showing the distribution of AUC values for boosted regression tree species distribution models using different numbers of trees. Models were fitted using the number of trees that optimized performance on the training data, up to 30,000 trees. The out-of-sample prediction performance (AUC) of models was generally unaffected by the number of trees used for models using more than about 2,000 trees.

Click here for additional data file.

Figure S12 Expected prediction performance of species distribution models (SDMs) for simulated species from the small community simulation under a range of sample size and spatial sampling bias scenarios

Panels show the expected prediction performance of logistic regression (GLM) and inverse distance-weighted interpolation SDMs. Lines show the predicted area under the receiver operating characteristic curve (AUC) from generalized additive models of prediction performance. Results from the small community simulation were qualitatively similar to results from the large community simulation.

Click here for additional data file.

We thank Tomás Murray and the Irish National Biodiversity Data Centre (NBDC) for providing and answering questions about the biological records data, and we thank the many citizen scientists who collected and contributed their data to the NBDC. This work used the ResearchIT Sonic cluster funded by UCD IT Services and the Research Office. This research used CORINE data made available with funding by the European Union.

Additional Information and Declarations

Competing Interests

Author Contributions

Data Availability

The authors declare there are no competing interests.

Willson Gaul conceived and designed the experiments, performed the experiments, analyzed the data, prepared figures and/or tables, authored or reviewed drafts of the paper, and approved the final draft.

Dinara Sadykova conceived and designed the experiments, performed the experiments, analyzed the data, authored or reviewed drafts of the paper, and approved the final draft.

Hannah J. White performed the experiments, authored or reviewed drafts of the paper, and approved the final draft.

Lupe Leon-Sanchez, Paul Caplat and Mark C. Emmerson conceived and designed the experiments, authored or reviewed drafts of the paper, and approved the final draft.

Jon M. Yearsley conceived and designed the experiments, performed the experiments, analyzed the data, authored or reviewed drafts of the paper, and approved the final draft.

The following information was supplied regarding data availability:

Code to run the simulation, analyze results, and produce all figures and tables are available on GitHub (https://github.com/wgaul/Data_quantity_is_more_important_than_its_spatial_bias_for_predictive_species_distribution_modelling) and from Zenodo: Willson Gaul. (2020, May 24). wgaul/Data_quantity_is_more_important_than_its_spatial_bias_ for_predictive_species_distribution_modelling: Preprint submission version (Version v1.1.0). Zenodo. http://doi.org/10.5281/zenodo.3841976.

Species data are publicly available from and must be accessed through the National Biodiversity Data Centre (https://www.biodiversityireland.ie/contact-us/) per the data permit requirements.

Bryophyte data are available from the NBN Atlas website (https://registry.nbnatlas.org/public/show/dr859 and https://registry.nbnatlas.org/public/show/dp74).

CORINE Land Cover data is available from https://www.eea.europa.eu/ds_resolveuid/ecb838dabf4849838ba5f3dc81ca6b0e.

E-OBS European Climate Assessment and Dataset data are available from http://www.ecad.eu/download/ensembles/downloadchunks.php.

Elevation data from the ETOPO1 Global Relief Model are available from https://www.ngdc.noaa.gov/mgg/global/relief/ETOPO1/data/ice_surface/grid_registered/netcdf/.

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
