# Peer review of "Data quantity is more important than its spatial bias for predictive species distribution modelling"

_PeerJ, doi:10.7717/peerj.10411_

## Round 0.1 · original submission · Major Revisions

Many thanks for submitting your manuscript to PeerJ. I personally very much enjoyed the manuscript however like one of the reviewers I did find it overly long and likewise I also got lost in the methods section. Also as the comparison is between presence only v what is presence/absence data it would be good if the text around it could be strengthened to show such a comparison is valid. However, I did enjoy this paper.

Reviewer 1 ·

Basic reporting

The authors present a really interesting paper comparing data quantity against spatial bias for predictive SDMs. With relevant findings and an innovative approach.

In general, the manuscript looks professional but some aspects can be improved:
L147-149. "Because our species were simulated... that species" needs a reference
L155, 160, 168, 171. Links and dates should be included in references list not in main text.
L337, 341, 347. Data and R code share should be included at the end of the document, usually links should appear at the end of the document or in the references.

Experimental design

In my opinion this research fits perfectly within Aims and Scope of the journal.
Research question is well defined, and the methodology is described with detail.

Validity of the findings

Results are cleared exposed, and can help to clarify the importance in data quantity and spatial bias in SDMs.
My only concern about the analysis is based in compare real data (only presences) with simulated data (presence/absence). I understand that some times it is not possible to obtain absence data. However, compare presences only against presence/absences should be done carefully. (just a reference to highlight the importance of the absences: The uncertain nature of absences and their importance in species distribution modelling
Lobo et al. 2010).

Reviewer 2 ·

Basic reporting

The basic structure of the article is adequate and the language clear. I believe readability would improve if some sections were shorter (introduction, methods) and clearer (methods; i.e. a step by step description and framework needed).

The references in the introduction fail to mention any virtual species research developed from 2017-2020, which is a clear problem.

Experimental design

The experimental design is adequate, but as previously mentioned, I believe the methods section can be clearer in some points. The amount of information and different processes/calculations leads the reader to get a bit "lost", so some clarification is needed.

Validity of the findings

I believe the findings have validity but the novelty needs to be better explained. However, even if not novel, this study is important because it reinforces the results obtained in previous studies, using different sample sizes and sampling strategies.

All the relevant data is available, but I really believe the manuscript will be improved if a framework picture was available.

Additional comments

Abstract

Line 8: I would say that the approach is novel. See for example https://www.sciencedirect.com/science/article/abs/pii/S1574954118301614 or https://besjournals.onlinelibrary.wiley.com/doi/full/10.1111/2041-210X.12203
Line 10: can the authors clarify what “strong” means?

Introduction
The introduction is well written, and the story makes sense. However, I believe that this section appears to be longer than necessary. Additionally, I think that the literature review that is needed in the introduction to tell the story until the moment of the study is lacking, since none of the references (or very few) are older than 2017. This is particularly true with the references for virtual species studies.
Line 65: can the authors provide some examples please?
Lines 91-95: it is important to reference that this study was performed with virtual species
Line 132: I would be more careful with the last sentence, especially since the most recent citation for relevant literature seems to be from 2016. What is a real-world spatial sampling bias for the authors? Using presences from real species to sample simulated ones? Please clarify.

Methods
This section is detailed. However, I believe it is longer than needed. It starts on page 11 and goes until page 20. Therefore, it is easy to lose yourself among so many details and calculations. I advise the authors to start the section with a small description of the entire procedure (e.g. step 1, step 2…) and to add a figure explaining the framework. I also find a bit confusing why the need to use real species data if the species were artificially created and randomly sampled. Please see my comments below but this real-virtual species connection needs to be clarified.

Lines 169 – 173: Does the type of data that was available is important (e.g. some insects, some birds)? Or the real important fact is that this is real data?
Lines 178 – 183: what would be the pros and cons of using a simulated sampling procedure instead of using real species data, beyond the obvious additional workload?
Lines 209-210: a community with 34 species and another one with 1268 means that in total 1302 species were created? Was this made to use match the real sampled points of real and virtual species (e.g. real species 1 data is the presence data of virtual species 1)? Not clear, so please clarify.
Additionally, why not create a large community and then just use a smaller subset of that data to test? Is it assumed that real data from insects and birds have different sampling bias? If this is true, what are the different bias? This is not very clear when describing the data.
Lines 221- 223: Ok, the authors reference my previous question here. But still not clear to me why they have different levels of sampling bias and how was this assessed?
Line 238 – 241: I apologize if I’m missing something but I don’t seem to understand why there is a need to associate sampled data from a virtual species to real bryophyte/dragonfly data, if the selection of points was done randomly? Perhaps clarify this please. I have the feeling that in these cases a framework picture is advisable.
Lines 261 – 267: If no pseudo-absences were used, does this mean that the authors used presence-absence data, or did the authors uses occupancy models?
Lines 268: if only 110 species were models, why create 1268 virtual species?
Line 295: Why AUC? I know this is a classic measure, but it can be misleading. Why not use MaxTSS or MaxKappa also, for example?

Results
Nothing really stands out in this section. All the relevant information/results are provided.
However, I really think that discussing results only based on AUC is not enough and other metrics should be used.

Discussion
This section is clear, and all the important points discussed. However, like in the introduction and methods, at some point I get the feeling that it is longer than needed and some structure required. I advise the authors to break this section in sub-sections wit clear themes being discussed in each.
I would like to see some recent references in this section also. I could only find one from 2018.
Line 412: Thibaud et al 2014 also used a random sampling approach. How different do the authors think this might be from the observed spatial sampling patterns
Line 431: can the authors suggest a way to “go around” the problem of using Ireland species and specific environmental variables, since virtual species were used?

Conclusions
I would like to see here some points giving ways forward in this area, or at least some points summarizing what is not advised to be done with SDMs, taking into account the research.

---

## Round 0.2 · accepted · Accept

The reviewer, like myself, very much appreciated the efforts you have gone to, to take on board the comments from the previous round. We believe it has made for a stronger and more readable paper that I am now happy to recommend be accepted for publication.

Reviewer 2 ·

Basic reporting

The structure of this manuscript was already good but I believe it was improved with the changes that the authors made to clarify some points. I also think that Fig.1 really helps the reader to have a better idea of the work developed. > Good Job!

Experimental design

Again, I believe only a few clarifications and edits were necessary regarding the explanation of the experimental design. The authors were receptive to my suggestions and edited some of the text to make it easier to understand, clarified some points that needed clarification and added some additional metrics to the analysis (i.e. Cohen's Kappa).

The division of the methods and Discussion sections in different steps was only a suggestion and by no means a demand (just a personal preference really). I'm happy with the changes made. Also, I feel the authors "pain" when fitting thousands of species and running simulations. > Well done

Validity of the findings

The authors clarified my point regarding the novelty of the work and I'm satisfied.

Additional comments

The authors did a good job addressing my comments and clarifying some doubts I had. I apologize to the authors if at times my questions seemed a bit strange but I think it helped to make the manuscript clearer at some crucial points.

I hope my comments were useful to the authors and that the final version of the manuscript is, in their minds, a better version of the initially submitted one. I certainly think it is.